# Scalable recombinase-based gene expression cascades

Tackhoon Kim[1,2], Benjamin Weinberg [3], Wilson Wong [3] & Timothy K. Lu [1✉]

Temporal modulation of the expression of multiple genes underlies complex complex biological phenomena. However, there are few scalable and generalizable gene circuit architectures for the programming of sequential genetic perturbations. Here, we describe a modular recombinase-based gene circuit architecture, comprising tandem gene perturbation cassettes (GPCs), that enables the sequential expression of multiple genes in a defined temporal order by alternating treatment with just two orthogonal ligands. We use tandem GPCs to sequentially express single-guide RNAs to encode transcriptional cascades that trigger the sequential accumulation of mutations. We build an all-in-one gene circuit that sequentially edits genomic loci, synchronizes cells at a specific stage within a gene expression cascade, and deletes itself for safety. Tandem GPCs offer a multi-tiered cellular programming tool for modeling multi-stage genetic changes, such as tumorigenesis and cellular differentiation.

[1] Research Lab of Electronics, Massachusetts Institute of Technology, Cambridge, MA, USA. [2] Chemical Kinomics Research Center, Korea Institute of Science and Technology, Seongbuk-gu, Seoul, Republic of Korea. [3] Department of Biomedical Engineering and Biological Design Center, Boston University, Boston, MA, USA. ✉email: timlu@mit.edu

Complex cellular tasks, such as those executed during normal development and tumorigenesis, are coordinated by multiple gene regulatory events operating across various time-scales. For example, the differentiation of cells into specific subtypes involves highly orchestrated transcriptional programs whose temporal regulation is controlled by transcriptional cascades of multiple genes[1]. Tumorigenesis involves the mutation of key tumor suppressors and proto-oncogenes, and different temporal orders of these mutations may change disease progression[2,3]. In a previous study, Clevers and colleagues sequentially delivered single-guide RNAs (sgRNAs) over a period of several weeks to organoids to model colorectal cancer[4]. However, this approach required strong positive selection for the cells in which the genes of interest had mutated[4]. A dual-recombinase system used to model sequential events in tumorigenesis in vivo[5] requires multiple independently integrated transgenes and may take months or even years to establish.

One of the central difficulties in programming gene expression cascades is scalability. Traditionally, cascades composed of multiple inducible gene expression systems require many distinct regulatory proteins that respond to different ligands (e.g., TetR for tetracycline)[6]. Thus, the number of constitutively expressed transgenes increases linearly with the number of independent genetic tasks in the gene expression cascade. Such cascades can be large, challenging to implement, and detrimental to cell viability because of resource competition at the transcriptional and translational level[7–9]. Furthermore, the number of well-validated and well-tolerated inducible systems available for use in mammalian cells and in animal models is limited.

Recombinases allow the robust and specific rearrangement of genetic elements. With a collection of highly orthogonal recombinases reported to be active in mammalian cells, these recombinases were successfully implemented for complex logical operations[10]. We have recently developed a library of >20 orthogonal split recombinases, enabling the independent regulation of expression of multiple transgenes using chemical ligands or light[11]. We envisioned that our collection of split recombinases could be used to overcome the scarcity of established inducible systems to drive gene expression cascades.

In this work, we overcome the limitations of previous gene cascades by designing an array of recombinase-based modules with memory that only require two distinct inducers; these modules reduce the constitutive expression of the transgenes needed to encode cascades. We leverage this cascade for the robust, sequential expression of sgRNAs to encode transcriptional cascades and trigger the sequential accumulation of mutations at endogenous genomic loci.

## Results

**Gene perturbation cassette (GPC): a recombinase-based gene expression module.** Each module performs two tasks: (1) the expression of payload gene(s), and (2) the self-termination of payload gene expression, followed by the expression of the next payload gene(s), in response to an inducer (Fig. 1a). To implement these features, we designed a compact, recombinase-based gene circuit called the GPC (Fig. 1b). The GPC is composed of three parts: (1) a split recombinase that is activated in response to chemically induced dimerization (CID) by gibberellin (GIB) or abscisic acid (ABA)[11–13], (2) payload gene(s) expressed in the same mRNA transcript as the split recombinase and terminator, and (3) recombinase recognition sites that flank all genetic elements of the GPC. Because a constitutively active promoter is placed upstream of the GPC, the split recombinase and payload gene are expressed until the cognate ligand activates the recombinase, which excises the entire cognate GPC. This terminates

recombinase activity and payload gene(s) expression and leads to expression of the next downstream GPC. This design represents an advancement in the scalability of programming gene expression cascades because temporal gene expression can be induced by simply alternating the exposure of the cells to just two ligands (ligand #1 for odd-numbered stages, and ligand #2 for even-numbered stages). The length of the gene expression cascade is limited only by the number of orthogonal recombinases; so far, nearly a dozen orthogonal recombinases have been confirmed to be active in mammalian cells[10]. Furthermore, this architecture further minimizes the cellular burden because only the recombinase and payload gene(s) in the proximal GPC are expressed at any given time.

Individual GPCs were optimized by implementing them in a simple two-stage cascade that switches expression from GFP (stage 1) to BFP (stage 2) upon ligand treatment and GPC excision (Fig. 1b). Split Cre, PhiC31, and Flp integrases were split at specific sites that give the highest signal-to-noise ratio, defined as the ratio of excision-induced BFP-positive cells in the presence and absence of CID ligand (Supplementary Fig. 1a). We then appended an additional nuclear localization signal to these proteins for added recombinase activity (Supplementary Fig. 1b). Furthermore, we chose Bovine Growth Hormone polyadenylation signal (BGHpA) as the most efficient polyadenylation signal to place at the end of each GPC in order to block leaky expression of downstream GPCs (Supplementary Fig. 1c). Optimized GPCs, including GIB-activated Cre (GIB-Cre), GIB-activated PhiC31 integrase (GIB-PhiC), and ABA-activated Flp (ABA-Flp), had signal-to-noise ratios of 80–792 in HEK293T cells (Fig. 1c, d). We confirmed that CIDs by GIB and ABA are orthogonal to each other (Fig. 1c, d). Payload gene switching from GFP to BFP was due to recombinase activity, as a GPC with an inactive mutant Cre (Cre Y324F)[14] failed to switch the expressed gene to BFP (Fig. 1c, d). Excision-dependent payload gene switching was confirmed by detection of an excision-specific PCR product (Fig. 1e).

**Tandem GPC enables robust gene expression cascade.** We next designed a gene expression cascade consisting of the tandem GPC array (Fig. 2a):

$$\text{GIB}-\text{Cre} \rightarrow \text{ABA}-\text{Flp} \rightarrow \text{GIB}-\text{PhiC}$$

Only the GPC directly proximal to the upstream promoter was expressed, consistent with our initial design in Fig. 1a. Thus, the initial GIB treatment, which activated GIB-Cre in the first GPC, did not excise the downstream GIB-PhiC GPC because the downstream GPCs were not initially expressed; this feature of the architecture allows GIB to be used again at stage 3 (GIB-PhiC GPC) within the same cascade. This tandem GPC gene circuit is self-sufficient in actuating a gene expression cascade and does not require any pre-integrated recombinase recognition sites. This feature, in principle, enables the tandem GPC to be rapidly implemented in any cell type of interest by simply integrating the gene circuit.

We next tested how quickly CID ligands can induce excision in a GPC. GIB treatment induced nearly complete excision of the GPC within 24 h; little additional excision occurred with longer durations of GIB treatment (Supplementary Fig. 2a). To allow for sufficient expression of the next gene and degradation of the previous gene in the cascade, we treated cells with CID ligand for 48 h for each stage in the cascade.

We also examined the rate at which ligands were cleared from the cells to determine the minimum time course required for switching between ligands. The GIB molecule most commonly

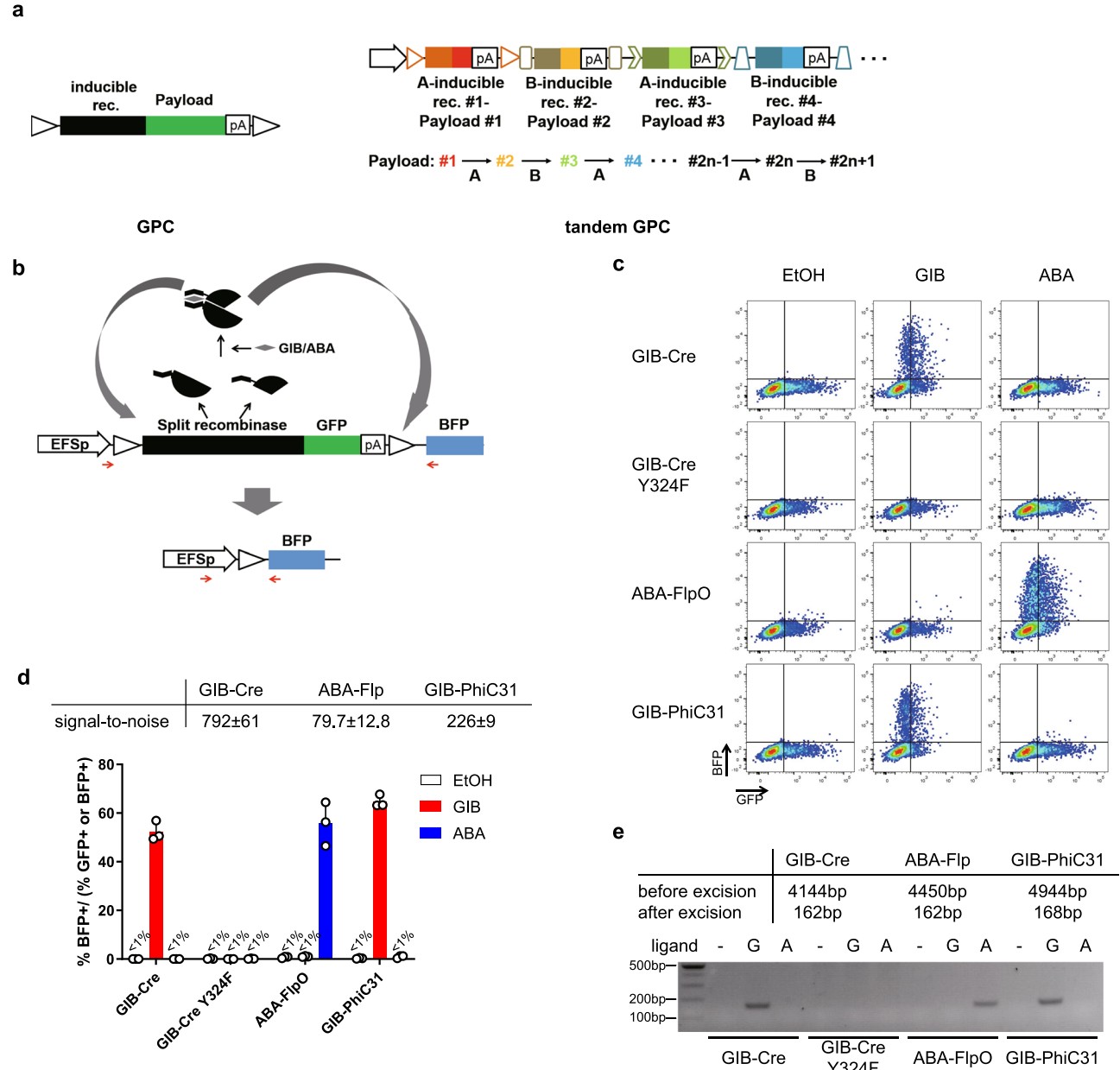

**Fig. 1 Design and validation of the gene perturbation cassette (GPC). a** Schematics of a single-unit GPC (left) and the tandem GPC circuit for programming gene expression cascades (right). **b** A single GPC enables the switching of the expressed gene from GFP to BFP. Red arrows indicate the primers used to detect the excision event in **e**. EFSp: human EF1a short promoter. **c** Representative flow cytometry plot for cells transfected with the single GPC architecture described in **b**, implemented with the indicated split recombinase and treated with the indicated ligands for 24 h. **d** Quantification of the results in **c** ($n = 3$, mean ± s.d.). %BFP+/(%BFP+ or GFP+) indicates the fraction of cells that had undergone recombination. **e** PCR validation of excision upon ligand treatment using primers depicted as red arrows in **b**. Source data are provided as a source data file. G: GIB, A: ABA.

used for CID in mammalian cells is acetoxymethyl group-modified gibberellin A3 (GA3-AM)[12], which can be trapped within cells by removal of the acetoxymethyl group by intracellular esterase. Other GIBs, such as GA4, are known to be yeast membrane permeable without any modification[15] and may readily diffuse out of the cells after ligand removal. As expected, GA3-AM required more than 12 h to be completely cleared from the cells whereas GA4 was immediately cleared from the cells after ligand removal (Supplementary Fig. 2b, c). Moreover, split-recombinase systems induced by GA4 were as efficient as those induced by GA3-AM (Supplementary Fig. 2c). We subsequently used GA4 to drive the gene expression cascade in the tandem GPC gene circuits throughout this study. We

similarly tested the kinetics of ABA clearance and found that the cells were virtually free of ABA 12 h after ligand removal (Supplementary Fig. 2c). Thus, we utilized a ligand treatment schedule of 48 h at each stage within the cascade, with a 12-h gap between switching of the ligands.

Excision by recombinases leaves a scar sequence consisting of recombinase-recognition sites (Supplementary Fig. 2d). Therefore, in our original tandem GPC (version 1), where the the start codon was placed within each GPC, sequential excision of the tandem array of GPCs inevitably leads to lengthening of the scar upstream of the GPC to be expressed. Longer scars in the 5′ untranslated region (5′UTR) may form secondary structures that affect translation efficiency[16]. To address this issue, we created a

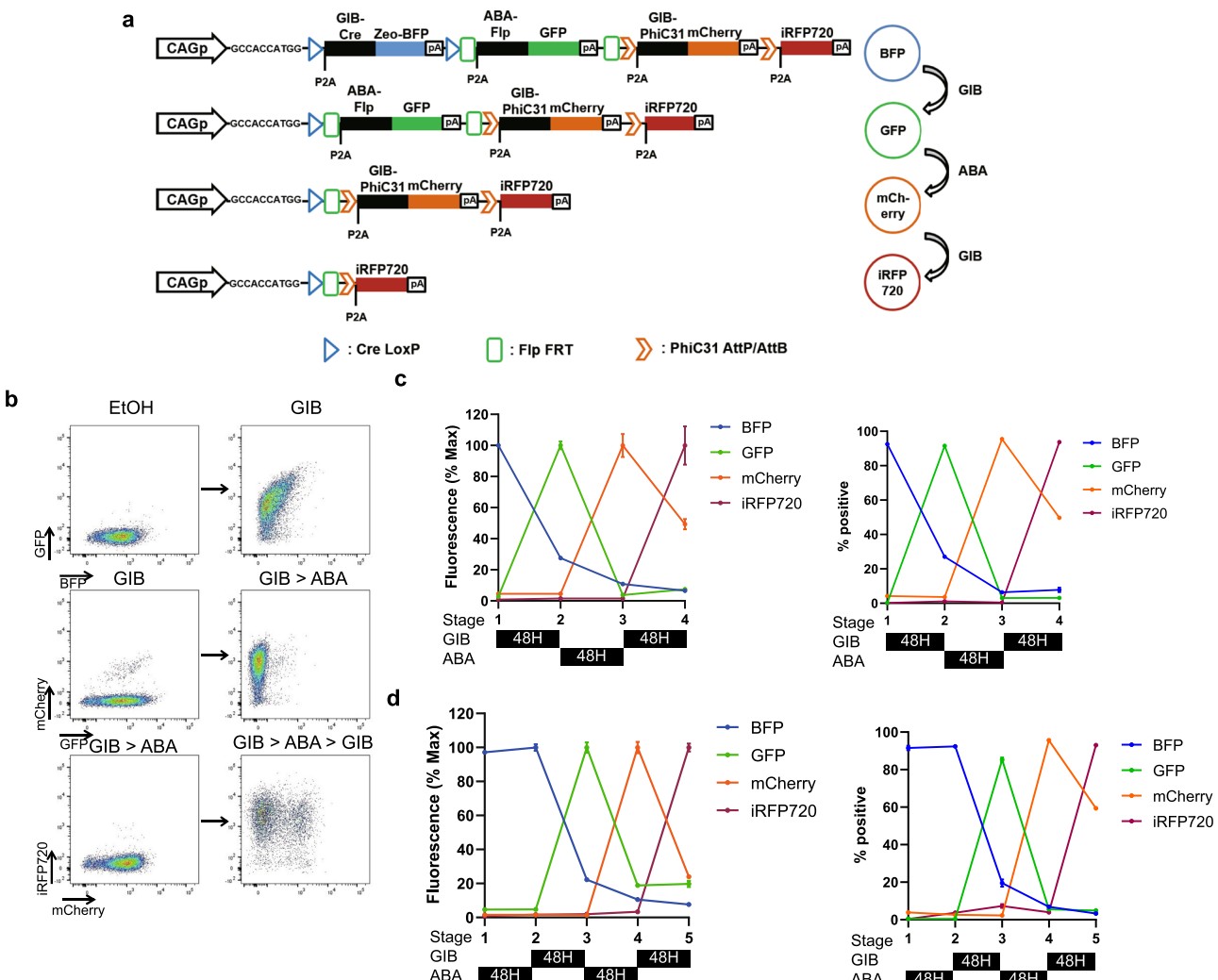

**Fig. 2 Design and validation of the tandem GPC gene circuit. a** Tandem GPCv2-Fluor gene circuit, used in **b–d**. Alternating treatments of GIB and ABA result in excision of GPCs and payload gene switching from BFP to GFP, to mCherry, and finally to iRFP720. Zeocin resistance gene is placed in the GIB-Cre GPC for positive selection of the gene circuit. CAGp: Hybrid promoter consisting of cytomegalovirus (CMV) early enhancer-chicken beta actin promoter-rabbit beta globin splice acceptor, P2A: porcine teschovirus-1 2A self-cleaving peptides. **b** Representative flow cytometry plot of HEK 293T cells carrying tandem GPCv2-Fluor treated with the indicated sequence of CID ligands. The flow cytometry plots for all relevant channels and gating schemes are provided in Supplementary Fig. 3a. **c** Quantification of mean fluorescence and average fraction of cells expressing the indicated fluorescent protein at each stage of the gene expression cascade ($n = 3$, mean ± s.d.). **d** Quantification of mean fluorescence and average fraction of cells expressing the indicated fluorescent protein after each indicated sequence of ligand treatment ($n = 3$, mean ± s.d.). The flow cytometry plots for all relevant channels and gating schemes are provided in Supplementary Fig. 3b. Source data are provided as a source data file.

version 2 tandem GPC (Supplementary Fig. 2e) with an initiation codon and a Kozak sequence between the promoter and the GPCs so that the scar sequence did not contribute to the 5′UTR but was actually translated. A self-cleaving 2A sequence was appended at the 5′ end of each split recombinase gene to reduce the possibility of the polypeptide from the scar sequence interfering with CID or recombinase activity. Consistent with our expectations, version 2 of the tandem GPC gene circuit had better excision efficiency than version 1, yielding a more robust gene expression cascade (Supplementary Fig. 2e). All gene expression cascades described hereafter were based on version 2 of the tandem GPC design.

We next tested tandem GPCv2-Fluor, a tandem GPC gene circuit that sequentially expresses four fluorescent proteins in HEK293T cells (Fig. 2a). Tandem GPCv2-Fluor consists of GIB-Cre-BFP, ABA-Flp-GFP, GIB-PhiC31-mCherry, and iRFP720[17] so that alternating treatment with GA4, ABA, and GA4 induces the sequential expression of BFP, GFP, mCherry, and iRFP720. All payload genes were tagged with the PEST sequence[18] to

increase their turnover rate so that fluorescence levels would more accurately reflect gene expression from each GPC. PiggyBac transposase was used to integrate large (>15 kb) gene circuits into mammalian host genomes[19]. Sequential treatment with GA4, ABA, and GA4, each for 48 h, induced the gene expression cascade

$$BFP \rightarrow GFP \rightarrow mCherry \rightarrow iRFP720$$

in up to 95% of the cells (Fig. 2b, c). We observed that a small (<10%) fraction of the cells expressed payload genes programmed to be expressed at later stages in the cascade. This expression was likely due to the leaky activity of split recombinases in the absence of ligand. Consistently, PCR amplification of the scar sequence detected a modest proportion of gene circuit at later stages of the cascade, indicating leaky recombinase activity (Supplementary Fig. 3b).

We further assessed the robustness of the gene expression cascade by changing the ligand treatment schedule or the order of

ligand treatment. We first tested whether cells harboring tandem GPCv2-Fluor maintained the memory of specific stages within the cascade. We did this by treating cells with GA$_4$ to switch the payload gene to GFP and then maintaining them in the absence of CID ligand for 8 days. More than 80% of the cells retained the correct memory of the stage in the cascade; that is, they preserved GFP expression and the capability to switch the payload gene to mCherry in response to ABA treatment (Supplementary Fig. 2f). We observed a minor (<10%) decrease in the percentage of GFP+ cells and premature mCherry expression in the absence of ABA. Point mutations that mitigate the tendency of split recombinase fragments to spontaneously reconstitute in the absence of CID ligand[20] are likely to further reduce leakage and improve the fidelity of memory. We also treated cells carrying a tandem GPCv2-Fluor with CID ligands in a different temporal order:

$$ABA \rightarrow GA_4 \rightarrow ABA \rightarrow GA_4$$

As expected, initial ABA treatment did not induce payload gene switching, nor did it disrupt the ability of tandem GPCv2-Fluor to execute the gene expression cascade upon subsequent GA$_4$ $\rightarrow$ ABA $\rightarrow$ GA$_4$ treatment (Fig. 2d).

**Tandem GPC enables sequential expression and mutagenesis.** Beyond expressing genes, tandem GPCs can be used for the sequential expression of sgRNAs to leverage the versatility of CRISPR-Cas9 for driving transcriptional programs (tandem GPCv2-CRISPRa) or sequential genome editing (tandem GPCv2-CRISPR). Payload genes in GPCs are expressed together with the recombinase from an RNA polymerase II promoter; this contrasts with the standard RNA polymerase III promoters used to drive sgRNA expression in most studies. Thus, we compared several strategies[21–24] for RNA polymerase II-driven sgRNA expression by flanking an sgRNA with RNA cleavage sequences using an assay in which the sgRNA targets dCas9-VPR to an artificial sgRNA-responsive promoter to induce mCherry expression (Fig. 3a, left)[25]. sgRNA flanked by 20nt core sequences for Csy4-mediated cleavage ("20nt core"), devoid of the 8nt "handle" sequence for Cas complex assembly[26], resulted in the strongest mCherry expression (Fig. 3a). Thus, in our tandem GPCv2-CRISPRa and tandem GPCv2-CRISPR circuits, the sgRNA was flanked by this 20nt core to induce the efficient liberation of sgRNA, while the remaining mRNA coding for the recombinase and lacking a poly-A tail was stabilized by appending a *MALAT1* lncRNA triple helix[27]. Each GPC had BGHpA appended at the 3′ end to prevent transcription of downstream GPCs.

We next implemented the optimized tandem GPCv2-CRISPRa, which activates the transcriptional cascade of endogenous genes *RHOXF2*, *ASCL1*, *HBG1*, and *TTN* upon sequential treatment with GA$_4$ $\rightarrow$ ABA $\rightarrow$ GA$_4$. We chose the genes in the transcriptional cascade to include those that have validated sgRNAs that efficiently bind and activate the promoters of interest[25,28]. We first examined the kinetics of decay of sgRNA after ligand treatment and excision of the GPC, and expression of the payload sgRNA in the next GPC. To this end, HEK293T cells carrying the tandem GPCv2-CRISPRa circuit were treated with GA$_4$. Then we measured the kinetics of expression of *RHOXF2* and *ASCL1*, which are the target genes of the sgRNAs expressed in the first and second GPC, respectively. In line with a previous report measuring sgRNA half-lives in cells[29], the expression levels of *RHOXF2* and *ASCL1* genes quickly changed, approaching equilibrium by 96 h (Supplementary Fig. 3). Therefore, we treated cells carrying tandem GPCv2-CRISPRa with CID ligands for 96 h to transition between each stage in this cascade. As expected, tandem GPCv2-CRISPRa actuated a transcriptional cascade of

*RHOXF2*, *ASCL1*, *HBG1*, and *TTN* in a temporally regulated manner (Fig. 3b, c).

We similarly implemented an sgRNA expression cascade for the sequential mutation of multiple genes by Cas9 to highlight the potential uses of this gene circuit for modeling the multi-stage nature of tumorigenesis[4]. To this end, we made tandem GPCv2-CRISPR, a circuit that expresses a cascade of sgRNAs to knock out key tumor suppressor genes in colorectal cancer. These genes, *APC*, *MLH1*, *SMAD4*, and *TP53*, are known to undergo sequential cancer-causing mutations[3]. A T7 endonuclease assay and next-generation sequencing analysis revealed that sequential indel mutations were triggered in these four genes in 54–89% of the cells in response to the proper sequence of inducers (Fig. 3d–f). We observed a modest lag in gene editing (e.g., *MLH1* editing between stages 2 and 3; *SMAD4* editing between stages 3 and 4). This lag was likely due to the sgRNA present at the time of excision. The sgRNA is expected to generate indel mutations until it gets degraded, therefore forming additional indel mutations after excision of GPC.

**Tandem GPC enables diverse sequential genetic events in one circuit.** The robustness of gene expression cascades is prone to decay as the number of stages within the cascade increases, because recombinases are not 100% efficient. Furthermore, removal of the gene circuit after completion of the cascade may be desired for safety. To address these points, we created the AttP-tandem GPCv2-CRISPR gene circuit to encode the gene expression cascade

$$sgAPC \rightarrow MLH1 \rightarrow Puro^R - sgSMAD4,$$

with AttP sites for PhiC31 integrase placed at the upstream end of the gene circuit (Fig. 4a). This all-in-one gene circuit sequentially generates indel mutations at *APC*, *MLH1*, and *SMAD4* loci and then, by puromycin selection, synchronizes the cells that are at a specific stage (in this case, stage 3) by removing any cells that are at other stages (in this case, stages 1 or 2) in the gene expression cascade (Fig. 4b). After synchronization, GA$_4$ treatment induces self-deletion of the AttP-tandem GPCv2-CRISPR gene circuit via recombination by PhiC31 integrase to prevent any further unwanted mutations. We confirmed the sequential accumulation of indel mutations in the sgRNA target loci (Fig. 4c, d). By quantifying the scar sequence flanking the recombinase recognition site of the proximal-most GPC for each stage (red arrow in Fig. 4a), we estimated that ~79% of the cells were synchronized at stage 3 in the gene expression cascade in response to puromycin selection (Fig. 4e). Synchronization removed cells that lacked indel mutations in the three loci due to inefficiencies in recombinase-mediated excision and subsequent gene expression cascade failure. The synchronization rate is likely an underestimate due to PCR bias that preferentially amplify shorter amplicon[30]. As the scar sequence gets longer after each stage (Supplementary Fig. 2d), when cells at different stages exist as a mixture, the shorter scar sequence, which corresponds to the earlier stage in cascade, will preferentially be amplified. Consistently, next generation sequencing of the equimolar mixture of the plasmids that have the scar sequence amplicon for each stage revealed that the shortest scar sequence amplicon (156 bp at stage 1) was overrepresented by 2.72–2.84 fold over other longer scar sequence amplicons (195 and 198 bp at stages 2 and 3, respectively) (Supplementary Fig. 5). Also, the presence of multiple copies of gene circuit in a cell can interfere with synchronization. As one copy of the stage 3 gene circuit that express puromycin resistance gene is sufficient to survive puromycin selection, puromycin does not eliminate the gene circuit

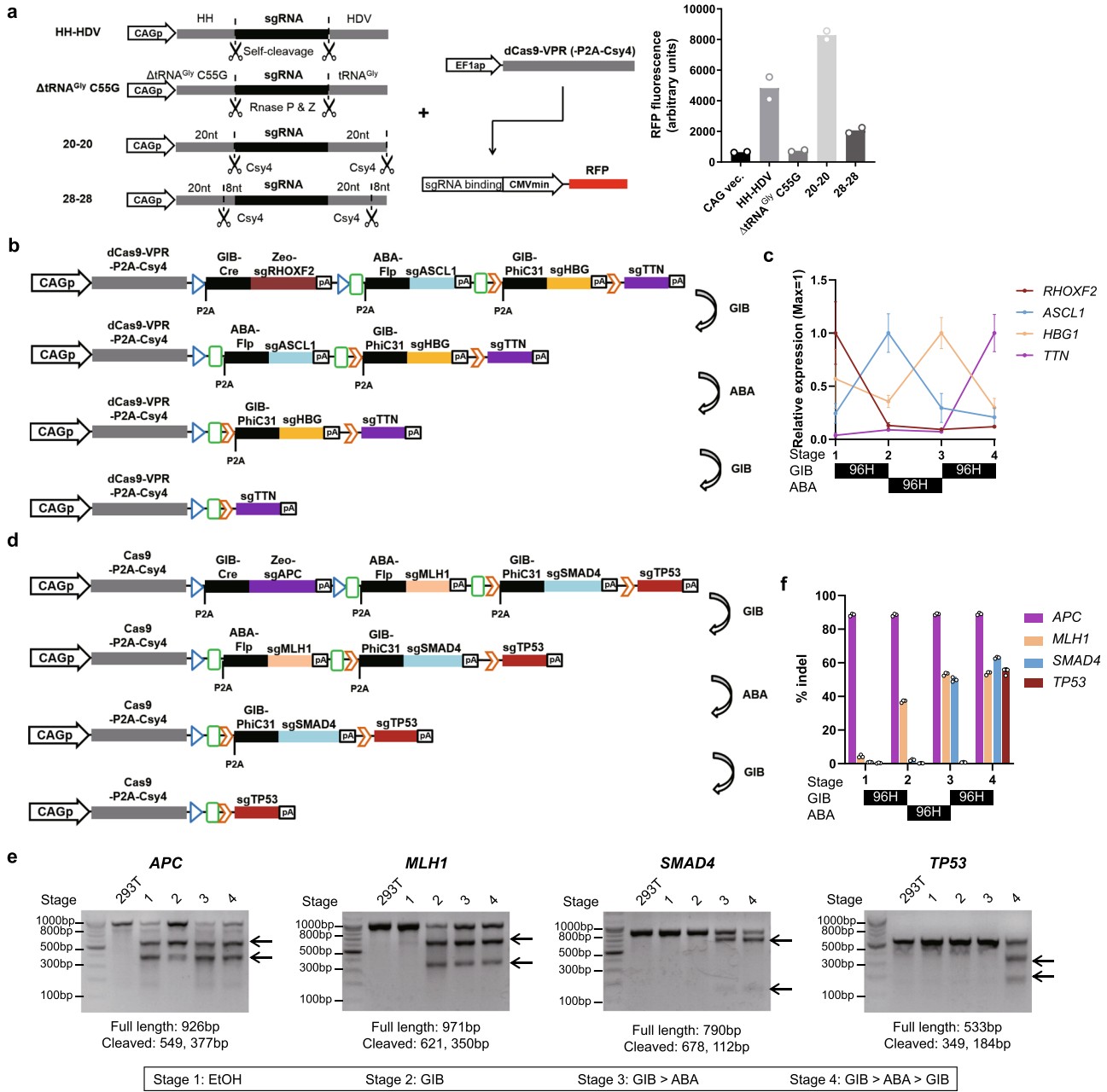

**Fig. 3 Sequential transcriptional cascade and gene editing by tandem-GPC-CRISPRa and tandem-GPC-CRISPR, respectively. a** (Left) Schematics of different architectures for RNA Pol II-driven sgRNA expression and reporter construct for sgRNA activity. (Right) Activity of RFP sgRNA reporter with indicated RNA Pol II-driven sgRNA format ($n = 2$, values indicate mean). EF1ap: human EF1a promoter, CMVmin: minmal CMV promoter. **b** Tandem GPCv2-CRISPRa schematics. **c** Quantification of sgRNA target genes after each indicated sequence of ligand treatment ($n = 3$, mean ± s.d.). **d** Tandem GPCv2-CRISPR schematics. **e** Validation of sequential gene mutations using T7 endonuclease assay. Arrows indicate cleaved bands. The expected lengths of uncleaved, and cleaved bands are noted below the gel image. **f** Quantification of gene editing in **e** by next generation sequencing (NGS) ($n = 3$, mean ± s. d.). Source data are provided as a source data file.

in other stages in cascade when it is in the same cell with at least one copy of stage 3 gene circuit. We have used very small amount of piggyback transposon vector (see the section "Methods") to limit copy number of the gene circuit to one copy per cell. Quantitative PCR analysis revealed that the average copy number of the gene circuit was very close to one (Supplementary Fig. 6). However, small fraction of cells that contain multiple copies of gene circuit may prevent perfect synchronization of the gene circuit.

The result of the synchronization was an increase in indel frequencies for *APC*, *MLH1*, and *SMAD4* genes from 28–62% to

46–84%. Finally, the activation of GIB-PhiC removed the entire gene circuit in 78% of the cells (Fig. 4f), leaving cells that had specific indel mutations but were devoid of any exogenous genes, including Cas9, being expressed in the gene circuit.

## Discussion

In summary, we demonstrated highly robust and scalable gene expression cascade circuits enabled by alternating treatment with two orthogonal ligands. We expect that gene circuits with diverse payloads in any arbitrary order will work in a "plug and play" manner, particularly if the flexibility of CRISPR-based

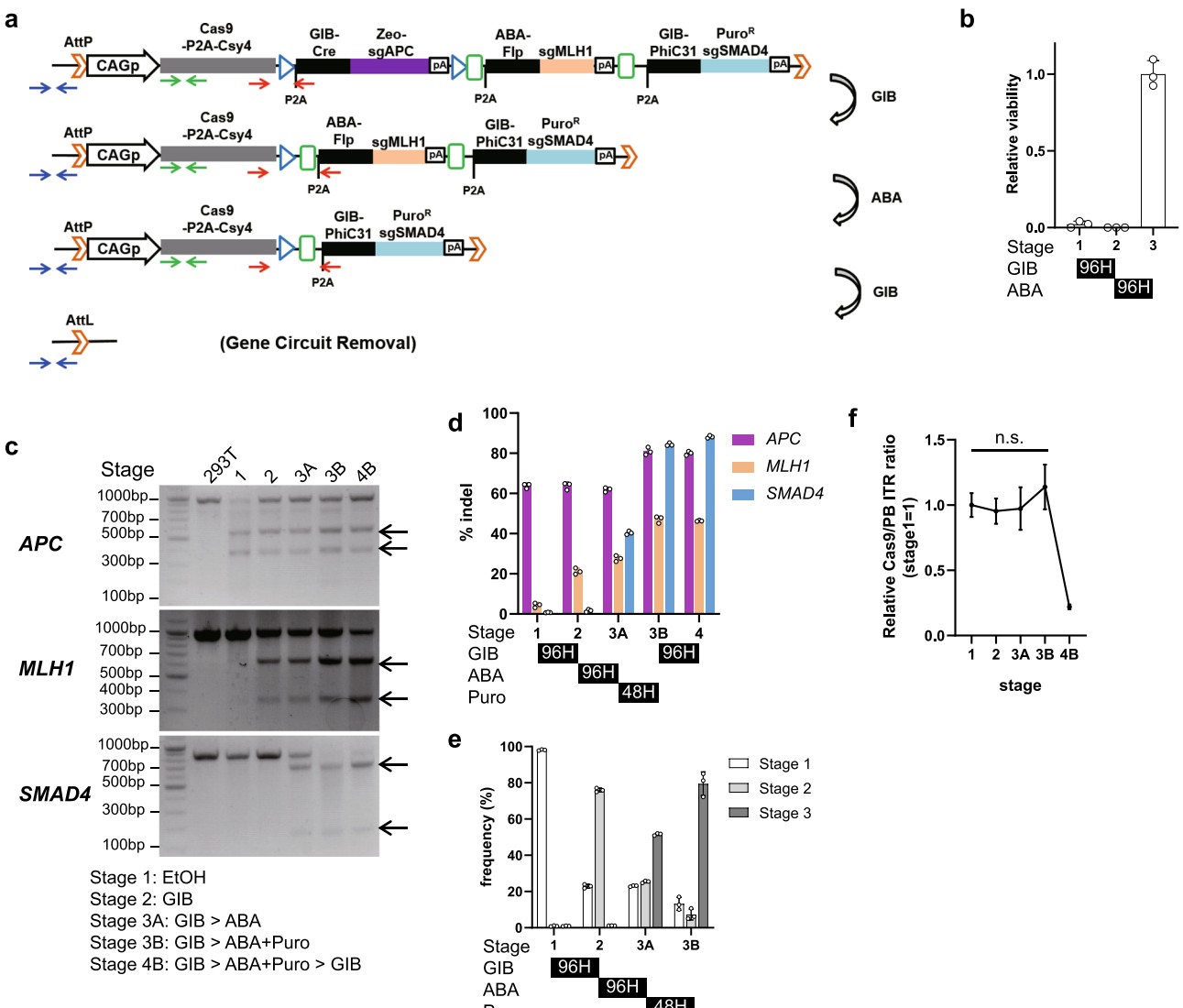

**Fig. 4 Sequential actuation, synchronization, and gene circuit removal by AttP-tandem GPCv2-CRISPR. a** AttP-tandem GPCv2-CRISPR schematics. Red arrows indicate primers used for amplifying the scar sequence to identify the stage within the cascade, used in **e**. Blue and green arrows indicate primers used for assessing gene circuit removal in **f**. Red arrows indicate primers used for amplifying scar sequence quantifed in **e**. **b** Relative viability of cells under puromycin selection at specific stages in the cascade ($n = 3$, mean ± s.d.). Data are normalized to the viability at stage 3. **c** Validation of sequential gene mutations using T7 endonuclease assay. Arrows indicate cleaved bands. The expected lengths of uncleaved and cleaved bands are the same as those in Fig. 3e. **d** Quantification of gene editing in **c** by NGS ($n = 3$, mean ± s.d.). **e** NGS quantification of frequency scar sequences that corresponds to each stage in the cascade ($n = 3$, mean ± s. d.). **f** qPCR validation of gene circuit removal assayed by relative amount of Cas9 gene in genomic DNA normalized to the inverted terminal repeat (ITR) of Piggybac transposon ($n = 3$, mean ± s.d.). Statistical analysis by two-tailed $t$ test. Source data are provided as a source data file.

transcriptional regulation and knockouts is leveraged. This feature thus enables the construction of gene expression cascades without intensive optimization.

Future work should optimize gene expression cascades so that there is complete synchronization of the cell population at each stage. To this end, we believe that it is critical to optimize split recombinases to achieve the highest on–off ratio in the presence and absence of CID ligands. The choice of split sites for recombinase most significantly influences the performance of split recombinases. Additional computational prediction of effective recombinase split sites[20,31], as well as empirical validation, will significantly improve gene expression cascades. Another major factor influencing the recombination efficiency is the expression level of the split recombinase. We have observed that cells with low GFP levels (hence expressing small amounts of split

recombinases) remained BFP negative (i.e., no recombination event) even in the presence of the CID ligand (Fig. 1c). Conversely, cells with high GFP (and split recombinase) expression managed to efficiently switch payload gene expression to BFP in the presence of CID ligand. Therefore, the use of efficient promoters to drive GPC expression is also critical for successful implementation of the gene expression cascade.

The presence of multiple copies of the tandem GPC gene circuit within a cell can undermine the fidelity of gene expression cascades, for several reasons. Recombination may occur between two copies of the tandem GPC gene circuit in the same cell, leading to inter-chromosomal translocations. Also, the multiple copies of the tandem GPC gene circuit may not be at the same stage within the cascade. The multiple copies of tandem GPC gene circuit at different stage thereby induces the cell to

simultaneously express multiple transgenes that are supposed to be expressed at different stages within the cascade. We have minimized the risk of multiple gene circuit integration by limiting the amount of transfected tandem GPC DNA[19]. And the average copy number of the gene circuit in the cells used in Fig. 4 were very close to one per cell (Supplementary Fig. 6). However, our current approach still has limits in precisely controlling gene circuit copy number. Even more precise control of copy number may be achievable with the use of a landing pad approach[32]. A landing pad approach would not only assist single-copy integration of the gene circuit but also allow the cascade to be expressed at uniform and predictable levels[32]. Our future work will involve integrating the gene circuit at a defined safe harbor genomic locus for improved fidelity of the gene expression cascade.

We anticipate that this gene circuit will be a useful tool for research and medicine. This circuit can be used to model cancer, both in vitro and in vivo, because it can be designed to closely mimic not only the multitude of mutations in human cancers but also the temporal order in which they occur. Similarly, we envision that this gene circuit can be applied for the efficient direct reprogramming of one cell type to another because it can be designed to reproduce the transcriptional cascades that occur in natural differentiation processes. This application should be useful in regenerative medicine to generate desired cell types.

## Methods

**Cell culture**. HEK 293T cells were obtained from American Type Cell Culture, and were cultured in Dulbecco's modified Eagle medium (Gibco) supplemented with 10% fetal bovine serum (Corning), and penicillin/streptomycin (Gibco).

**GPC excision test**. HEK 293T cells were seeded in 96-well plates, and transfected 12–16 h after seeding with unit GPC plasmid using Fugene HD (Promega). Ligands for CID are treated starting 6 h after transfection for 24 h. Cells were trypsinized for flow cytometry analysis using LSR Fortessa (BD Biosciences). The efficiency of excision was calculated as (% BFP positive)/(%BFP or GFP positive) to account for transfection efficiency.

**Gene circuit integration with Piggybac transposase**. HEK 293T cells plated in six-well plate was transfected with mixture of 2 μg empty plasmid + 50 ng Piggybac transposon plasmid containing the gene circuit + 25 ng Super Piggybac transposase (System Biosciences). Cells that successfully integrated the gene circuit was selected with 1 mg/mL zeocin.

**Flow cytometry**. Data obtained with LSR Fortessa (BD Biosciences) were compensated for spectral overlap and analyzed with FlowJo (FlowJo LLC.).

**Ligand decay assay**. HEK 293T cells were seeded in 96-well plates, and pretreated with CID ligands (GA$_3$-AM[Toronto Research Chemicals], GA$_3$ [Sigma], GA$_4$ [Sigma], ABA [Gold Biotechnology]). Twenty-four hours later, cells were washed with phosphate buffered saline and trypsinized for seeding in new 96-well plates in the absence of ligands. For zero hour wash, cells were seeded with DNA–Fugene HD mixture for reverse transfection. For other washout time points, cells washed out of ligands were trypsinized and seeded 12 h before transfection. GIB-Cre and ABA-FlpO GPCs are transfected for GIB and ABA decay assays, respectively.

**DNA constructs**. Unit GPCs with optimized split recombinases are digested with PacI restriction enzyme (New England Biolabs), and was used to perform Gibson assembly with PCR amplicon containing the payload gene, 3× bovine growth hormone polyadenylation sequence, and 2× chicken hypersensitivity site 4 core sequence. The resulting unit GPCs were digested with SapI restriction enzyme (New England Biolabs) for golden gate assembly with SapI restriction enzyme and T4 DNA ligase (New England Biolabs). The complete list of plasmids used in this study is in Supplementary Table 1. The protospacer sequences for sgRNAs are listed in Supplementary Table 2.

**Reverse transcription and quantitative PCR**. Total RNA from the cells were isolated with Trizol (Life Technologies) according to the manufacturer's instructions. The resulting RNAs were reverse transcribed with MMLV reverse transcriptase (Promega). Complementary DNA were used as templates for quantification of genes using TOPreal qPCR 2× Premix (Enzynomics), and Applied Biosystems 7500 real-time PCR system. Primers used for quantitative PCR are listed in Supplementary Table 3.

**T7 endonuclease assay**. Genomic DNA of HEK 293T cells were isolated using DNeasy Blood & Tissue Kit (Qiagen). Genomic loci flanking the sgRNA target sites were PCR amplified with Q5 DNA polymerase (New England Biolabs). The PCR product was purified with gel DNA recovery kit (Zymo Research Corp.). The purified PCR product was denatured at 95 °C for 5 min, and annealed by slowly cooling with ramp speed of −0.1 °C/s to 25 °C. The annealed PCR product was digested with 5U T7 endonuclease I (New England Biolabs) for 30 min in 37 °C for gel electrophoresis. Primers used for amplifying sgRNA target genomic loci are listed in Supplementary Table 4. The uncropped, unprocessed gel images are in source data file.

**Next generation sequencing**. The PCR amplification product of size 200–300 base pairs flanking the sgRNA target sites sites or that of size 150–200 base pairs flanking the scar sequence of the gene circuit were purified with gel DNA recovery kit (Zymo Research Corp.). The DNA library was prepared with Illumina TruSeq Nano DNA library Construction (insert size 350 bp). The resulting DNA library was sequenced with HiSeq4000 (Illumina, 150nt paired-end). The next generation sequencing data was analyzed for indel frequency using CRISPRESSO2[33]. The primers used for generating amplicon for NGS analysis are listed in Supplementary Table 5.

**Quantitative PCR analysis of gene circuit excision**. Genomic DNA of the cells with gene circuit were harvested with DNeasy Blood and Tissue Kit (Qiagen). Cas9 gene and the inverted terminal repeat (ITR) of the Piggybac transposon harboring the gene circuit are measured by quantitative PCR. Relative amount of Cas9 gene normalized to that of ITR was calculated to assess gene circuit excision efficiency. The primers for amplifying Cas9 and ITR are listed in Supplementary Table 6.

**Quantitative PCR analysis of gene circuit copy number**. A 145 base pairs amplicon at human GAPDH genomic DNA locus was subcloned into a Piggybac transposon vector. This plasmid is used as a reference DNA with Piggybac ITR DNA and GAPDH DNA ratio of 1:1. GAPDH and ITR DNA content was quantified from genomic DNA from cells with the tandem GPC gene circuit using quantitative PCR. The relative amount of ITR DNA compared to the reference plasmid was calculated to estimate copy number of the gene circuit. The primers for amplifying human GAPDH genomic fragment are listed in Supplementary Table 6.

**Reporting summary**. Further information on research design is available in the Nature Research Reporting Summary linked to this article.

## Data availability
The plasmids are available upon request. The NGS data is deposited in Short Read Archive (SRA) PRJNA680170: SRR13106981, SRR13106982. The custom codes used for analyzing NGS data are deposited in github: https://github.com/tackhoonkim/GPC-NatComms2021. Source data are provided with this paper.

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

## Acknowledgements

We thank Nathaniel Roquet for helpful discussions. This work was supported by Korea Institute of Science and Technology (KIST) Institutional Programs (2E30240 to T.K.); Human Frontier Science Program (LT000595/2017-L to T.K.), National Research Foundation of Republic of Korea (2016R1A6A3A03011376 to T.K.) and the Department of Defense (LC170525 W81XWH-18-1-0513 to T.K.L.)

## Author contributions

T.K., T.K.L. conceived the concept. T.K., T.K.L., B.W. and W.W. wrote the manuscript. T.K. performed all research. B.W. and W.W. provided the split recombinases used in the research. T.K.L. supervised the research.

## Competing interests

T.K.L. is a co-founder of Senti Biosciences, Synlogic, Engine Biosciences, Tango Therapeutics, Corvium, BiomX, and Eligo Biosciences. T.K.L. also holds financial interests in nest.bio, Ampliphi, IndieBio, MedicusTek, Quark Biosciences, and Personal Genomics. Other authors declare no competing interests.

## Additional information

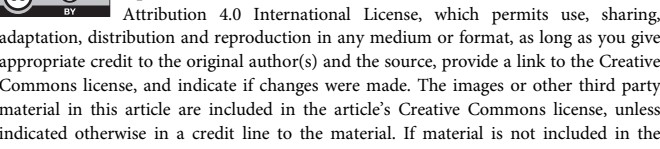

