## [Peer Review File · Nature Communications]

Reviewers' Comments:

Reviewer #1:

Remarks to the Author:

In the manuscript, Kim et al. described a scalable and modular recombinase-based gene circuit (gene perturbation cassettes, GPC) architecture to allow expression of multiple genes in a defined sequential order in response to two orthogonal ligands (GIB and ABA). The authors used tandem GPC to express up to 4 sgRNAs for sequential transcriptional activation and gene editing. In addition, the authors synchronize cells at a specific stage by using selection marker gene (PuroR) and delete the remaining GPC cascade by using inducible split recombinase. The manuscript is well-written and has potential implementations in scenarios that are required multi-stage cellular programming and genetic modifications. A few issues are necessary to be addressed to improve the quality of this manuscript.

1. In Fig. 2d, the excision efficiency is about 50~70% for three split recombinases, which limits the potential real applications. It is worth discussing the reason for the low recombination efficiency in HEK293 cells and how to further improve recombination efficiency.
2. In Fig. 2b, leaky mCherry and iRFP720 expressions are observed at the stage of GIB and GIB>ABA. Is this caused by overriding the terminator sequences? If this is the reason, introducing premature stop codons might reduce the leaky expression. And it is necessary to show the FACS results of all four fluorescent channels at each stage to identify the leaky expression.
3. In Fig. 4, it is not clear to me that time interval between each stage. In Fig. 4e, the recombination efficiency is only about 75% which is lower than the >90% in Fig 2. What causes this discrepancy? Is there any unexpected recombination event detected by NGS? Ideally, the puromycin selection can eliminate all negative cells in stage 3B. Can authors explain why ~10% and 5% of cells in stage 1 and 2 survived after the puromycin selection.
4. Why a MALAT1 lncRNA triple helix was used instead of poly-A signal in Fig. 3?

Reviewer #2:

Remarks to the Author:

Review of manuscript: “Scalable recombinase-based gene expression cascades”

Senior Author: Timothy K. Lu

Journal considering: Nature Communication

Summary:

The main focus of this manuscript is to build and test recombinase-based gene circuits comprising tandem gene perturbation cassettes (GPCs) that enable the sequential expression of multiple genes by alternating between treatment with two orthogonal ligands, GIB and ABA. With this technology, they can trigger sequential accumulation of mutations, sequentially express CRISPR-Cas sgRNAs, and synchronize cells at specific stages within a gene expression cascade. Finally, they were able to build a single circuit capable of sequential edits to a specific locus in the genome, synchronize the cells to a specific stage of the gene expression cascade, then have the circuit delete itself for improved safety considerations.

Major Comments:

1. Overall, the manuscript is well written, however, the authors claim that these circuits can be used to study complex cellular events such as tumorigenesis and differentiation, yet they do not show any data that supports these claims. The specific concern is related to the timing of the system. For example, the temporal switching described by this system seems to require a ligand treatment schedule of 48 hours per ligand treatment, with an additional 12-hour gap between switching ligands, in order to achieve optimal gene expression. This temporal limitation would not allow for gene expression cascades that need to occur on a faster timescale for most translational applications. The example brought up in the introduction by the authors (with tumorigenesis) occurs on a timescale where gene expression changes within 12-hour increments (DOI: 10.1093/carcin/bgt480). Any desired form of therapeutic delivery that one would want expressed at a certain cascade stage could not be implemented using these circuits due to the inherent temporal limitation. One suggestion is to find a different biological example that operates on the timescale that the circuit does.
2. The cell line used is not described. Do the cell lines have the recombinase recognition sites stably integrated into them?
3. The gene circuit schematics have several unlabeled features that are not annotated on the figure itself, or in the figure legends (e.g. recombinase sequences, promoters, etc.). Labeling these features and noting them in the legend will help the broader readership. Additionally, color code recognition sites in the schematics so that they match the genes which encode their recombinase. Finally, consider showing multiple steps in a schematic how each GPC is cleaved sequentially leading to activation of the next gene.
4. The notation of the T in the schematics of the circuits is confusing. Typically, this is used as a transcriptional terminator in prokaryotic circuits (sequence that mark the end of transcription), but the authors use it to indicate the polyA tail. It is unclear why this needs special annotation – polyA tails provide stability to the RNA molecule and don't impact transcription. If the authors want to keep it in their schematics, it would be prudent to explain why it is necessary.
5. In figure 1b it would be useful to know the size between the red primers, especially since data is collected to determine whether the space between these primers is removed after recombination.

6. The flow cytometry data in figure 1c is missing numbers on the axes. Despite being arbitrary values, it is always informative to know the difference between the lowest and highest values. Are the axes linear? Log scale?
7. The y-axes on these plots should be altered so the spread in the data around zero is larger. This can be done with flow data in FlowJo by changing the y-axis values to biexponential, then changing the width basis. This can also be done for Figure 2b and the flow data in the Extended Data.
8. Figure 1d shows a quantification of the flow cytometry data presented in 1c, however without a description of what cells are being looked at in the figure legend, this figure is a little confusing.
9. Figure 1e is not informative at all. Without knowing what size of the fragments before excision and no ladder present on the gel, this gel could be just about anything.
10. The same comments made for figure 1c apply to figure 2b.
11. In 2C and 2D, the pie charts are quite large and are distracting (and they all look the same). Instead of the pie charts, consider preparing a time course graph (like was done with mean fluorescence) showing the average fraction of cells at each time point expressing each of the 4 fluorescent factors.
12. In figure 3B the authors swap out fluorescence reporter genes with other genes. It is never discussed why these particular genes were chosen. Indeed, in the main text it describes using the circuit for a cascade of endogenous genes RHOXF2, ASCL1, HBG1, and TTN, however, there is no indication that these have any natural relationship *in vivo*. It would be helpful if the authors could provide their reasoning for choosing these genes in particular to be expressed sequentially.
13. Figure 3d should have an associated line graph, much like Figure 3c.
14. The gel in figure 3E needs a ladder. Also, what are the arrows supposed to be pointing to? This should be in the figure legend.
15. Figure 3f is not clear. A better explanation of this data in the legend and text would be very helpful. Why is APC expressed in all four stages? Why is there a lag in the data for MLH1 between stage 2 and stage 3? Finally, please consider displaying the data with the stages on the x-axis and use color to label the genes. This seems more intuitive. The same suggestion is for 4d, and 4e (as shown below).

16. In Figure 4, the activation of GIB-PhiC removed the entire gene circuit in only 67% of the cells, leaving a fairly substantial 33% of cells with the circuit intact. This level of efficiency could certainly pose issues in any future translational work, if the goal was to completely remove the entire circuit. It would be nice if the authors could provide potential ways of improving the design to improve the removal of the entire circuit at the end, perhaps in the final concluding paragraph.

17. In 4b. Stage 3=1 does not make sense. Just say in the figure legend that you normalized to stage 3.

18. How is the data in figure 4f analyzed? Delta delta Ct? Since it's relative gene content on the y-axis, it's unclear what exactly is happening in this figure and the legend has no information on this. It also doesn't make sense that the circuit content increases going from stage 1 to 2. How can the circuit content increase with a removal of DNA? More description on this data would be very helpful.

Minor comments:

1. Some full-forms of acronyms should be mentioned in the main text or in the figure legend before using it in the figures. For example: BFP, EFSp as used in Fig. 1b, CAGp as used in Fig. 2a, p2A.

2. Since the entire paper uses split recombinase to a great extent, it is suggested that it is explained better to explain better in the introduction, along with its general mode of action of recombinase in a few lines.

3. Extended Fig. 1c needs better explanation. What does the white bar stand for? They also did not mention the polyadenylation signal that was finally selected after comparison.

One suggested experiment is to test ligand treatment schedule for 24 hours and 36 hours with a 12-hour gap between switching of ligands. This would enable them to demonstrate the comparison of results between 24, 36 and 48 hours.

4. In Fig. 1b, a split recombinase site can be shown in a black rectangle near BFP just like it has been shown for GFP for clarity of the concept

5. Line 103: The word 'ligand treatment' should be removed to avoid repetition and for sentence accuracy.

6. The colour code of the genes shown in Fig. 3b and Fig. 3d can be changed from those used for proteins shown in Fig. 2a to avoid confusion among readers
7. Line 172: There is an error in the spelling of the word 'knockout'.
8. What does the x-axis on the graphs of Extended Fig. 1a stand for?

We thank the reviewers for helpful comments. Overall, we have added more discussion of how the performance of our gene expression cascade may be improved. The figures have been modified to provide better support for our scientific claims. Our point-by-point responses to the reviewers' points are written in blue.

Reviewer #1 (Remarks to the Author):

In the manuscript, Kim et al. described a scalable and modular recombinase-based gene circuit (gene perturbation cassettes, GPC) architecture to allow expression of multiple genes in a defined sequential order in response to two orthogonal ligands (GIB and ABA). The authors used tandem GPC to express up to 4 sgRNAs for sequential transcriptional activation and gene editing. In addition, the authors synchronize cells at a specific stage by using selection marker gene (PuroR) and delete the remaining GPC cascade by using inducible split recombinase. The manuscript is well-written and has potential implementations in scenarios that are required multi-stage cellular programming and genetic modifications. A few issues are necessary to be addressed to improve the quality of this manuscript.

1. In Fig. 2d, the excision efficiency is about 50~70% for three split recombinases, which limits the potential real applications. It is worth discussing the reason for the low recombination efficiency in HEK293 cells and how to further improve recombination efficiency.

We welcome the reviewer's comments. We believe there are two key factors limiting the recombination efficiency: (1) the expression level of the split recombinases; (2) the optimized split sites for the split recombinases. As seen in figure 1c, the cells negative for BFP (thus failed to undergo recombination), had low GFP expression level (thus low split recombinase expression level). Therefore, only the cells that express a sufficient amount of recombinase managed to undergo recombination. Also we have shown in supplementary figure 1a that the choice of split site for split recombinases is critical for optimal activity of the recombinases. Future work should optimize split recombinases by computationally predicting effective split site for recombinase and empirically validating the predictions (Dolberg, biorXiv, 2019, Daliyan Nat. Comms. 2018). We newly added the discussion on optimizing the GPC efficiency in the discussion section (lines 264-277).

2. In Fig. 2b, leaky mCherry and iRFP720 expressions are observed at the stage of GIB and GIB>ABA. Is this caused by overriding the terminator sequences? If this is the reason, introducing premature stop codons might reduce the leaky expression. And it is necessary to show the FACS results of all four fluorescent channels at each stage to identify the leaky expression.

We appreciate the reviewer's detailed concern. There was minor but detectable leaky expression of the mCherry and iRFP720 genes before the cells reached the stage at which the genes are supposed to be expressed. We do not think the leaky downstream GPC expression is due to overridden terminator sequences. Each GPC already has transgenes with multiple stop codons at the 3' end, as the reviewer had suggested, so while it is possible that downstream GPC can be transcribed with overridden terminator sequences, it would be very unlikely for the genetic contents of the downstream GPC to be translated because the polypeptide chain would have been terminated in the upstream GPC.

We think that the leaky recombinase activity in the absence of CID ligand is responsible for leaky expression of the downstream GPCs. To demonstrate this, we PCR amplified the "scar sequence" that can specifically amplify small fragment of the gene circuit flanking the recombinase recognition sites (described in Supplementary figure 2d). We performed NGS analysis of the scar sequence amplicon. In newly added Supplementary Fig. 3b, we detected modest but reproducible amplification of the "scar sequence" that corresponds to the stages later than the intended stage of the cascade (e.g., scar sequence corresponding to stage 3 detected at stage 2). This indicates that premature recombination of the GPC had occurred in the absence of any CID ligands, causing leaky expression of downstream GPCs.

In addition, as suggested by the reviewer, we newly added the flow cytometry plots for all four fluorescence channels for Fig. 2c, d in Supplementary Fig. 3a and c, respectively.

3. In Fig. 4, it is not clear to me that time interval between each stage. In Fig. 4e, the recombination efficiency is only about 75% which is lower than the >90% in Fig 2. What causes this discrepancy? Is there any unexpected recombination event detected by NGS? Ideally, the puromycin selection can eliminate all negative cells in stage 3B. Can authors explain why ~10% and 5% of cells in stage 1 and 2 survived after the puromycin selection.

We thank the reviewer for clarifying our manuscript. We added the drug treatment schedule under the graphs in figures 4d, and 4e to better clarify the time interval.

We did not observe any unexpected recombination events in our NGS data. We believe one of the reasons for the discrepancy between our data in Fig. 4e and Fig. 2 is the PCR bias during the amplification of scar sequences. The method we used to quantify the cells at a specific stage within the cascade was measured by next generation sequencing of the scar sequence, which contains a distinct set of recombinase recognition sites at each stage. We used a single pair of primers to amplify all types of the scar sequence from the genomic DNA of the pool of cells at each stage. As noted in Supplementary Fig. 2d, the scar sequence gets longer after each excision due to the accumulation of recombinase recognition sites. When the scar sequences of varying lengths are present in a mixture of cells, smaller amplicons tend to be more efficiently amplified. Therefore, we reasoned that NGS quantification of the scar sequence underestimates the excision frequency. This is particularly evident when we compared the results in Fig. 4d and 4e. While the cells at stage 3 by scar sequence quantification reached ~75%, the indel frequency of SMAD4 gene, which can only be edited at stage 3, exceeded 80%. This implies that the fraction of cells at stage 3 is actually at least higher than 80%. To provide evidence for the PCR bias, we performed next generation sequencing analysis of the PCR amplicons from an equimolar mixture of the three scar sequences corresponding to stage 1~3 of the cascade. Consistent with our expectations, the shorter amplicon (156bp) that corresponds to stage 1 is over-represented by 2.72~2.84 fold compared to the longer amplicons (195bp, 198bp) that correspond to stages 2 and 3 (Supplementary Fig. 6). PCR bias thus over-represents the cells at stage 1, and therefore underestimates the gene expression cascade efficiency. The new results and discussion regarding the PCR bias in the analysis have been added in lines 239-246.

Another factor that limits robustness of the gene expression cascade is the copy number of the gene circuit. We have used the Piggybac transposon to deliver our gene circuit. If the transposon vector is used at low enough concentration, the copy number in most cells can be kept at one (Li, Nucleic Acids Research, 2011). However, there could be a small fraction of cells that had integrated multiple copies of the gene circuit. As one copy of the gene circuit at stage 3 would suffice for cells to survive puromycin selection, cells with multiple copies of the gene circuit that have at least one copy of gene circuit at stage 3 may retain gene circuits that are at stage 1 or 2. If this occurs, while every single cell would have at least one copy of the gene circuit synchronized at stage 3, not all gene circuits would be synchronized. We examined the copy number of our gene circuit with qPCR analysis. As HEK293T cells are largely triploid across all chromosomes, one copy of transgene corresponds to 1/3 of the amount of DNA for any given endogenous locus. The amount of the gene circuit relative to the GAPDH locus was found to be ~0.3 (Supplementary Fig. 5), indicating that the average number of gene circuits per cell is close to 1. However, it is possible that there is more than one copy of the gene circuit in a small fraction of the cells. The new results and discussions regarding the copy number of transgene are added in lines 246-254, 278-293. We hope this clarifies the reviewer's concern.

4. Why a MALAT1 lncRNA triple helix was used instead of poly-A signal in Fig. 3?

Both MALAT1 lncRNA triple helix and poly-A signal were used for all the gene circuits described in Figures 3 and 4. We clarified this in our revised manuscript (line 194-195). We apologize for any confusion this may have caused.

Reviewer #2 (Remarks to the Author)

Summary:

The main focus of this manuscript is to build and test recombinase-based gene circuits comprising tandem gene perturbation cassettes (GPCs) that enable the sequential expression of multiple genes by alternating between treatment with two orthogonal ligands, GIB and ABA. With this technology, they can trigger sequential accumulation of mutations, sequentially express CRISPRCas sgRNAs, and synchronize cells at specific stages within a gene expression cascade. Finally, they were able to

build a single circuit capable of sequential edits to a specific locus in the genome, synchronize the cells to a specific stage of the gene expression cascade, then have the circuit delete itself for improved safety considerations.

Major Comments:

1. Overall, the manuscript is well written, however, the authors claim that these circuits can be used to study complex cellular events such as tumorigenesis and differentiation, yet they do not show any data that supports these claims. The specific concern is related to the timing of the system. For example, the temporal switching described by this system seems to require a ligand treatment schedule of 48 hours per ligand treatment, with an additional 12-hour gap between switching ligands, in order to achieve optimal gene expression. This temporal limitation would not allow for gene expression cascades that need to occur on a faster timescale for most translational applications. The example brought up in the introduction by the authors (with tumorigenesis) occurs on a timescale where gene expression changes within 12-hour increments (DOI: 10.1093/carcin/bgt480). Any desired form of therapeutic delivery that one would want expressed at a certain cascade stage could not be implemented using these circuits due to the inherent temporal limitation. One suggestion is to find a different biological example that operates on the timescale that the circuit does.

We thank the reviewer for raising this concern. We have now more thoroughly explained evidence that our gene circuit can be implemented in modeling tumorigenesis in lines 42-48. The actual tumorigenesis observed in clinic involves a gradual accumulation of causative mutations in the time scale of years. Accordingly, previous approaches, both in vitro using organoid cultures and in vivo using mouse models, have involved the sequential introduction of gene mutations over a period of weeks or months. While we agree that our gene circuit has limitations to be implemented for the sequential gene expression changes or gene mutations that occur in several hours, we believe that our gene expression cascade can reliably be used to model life phenomena that occur in the period of days, weeks, or months.

2. The cell line used is not described. Do the cell lines have the recombinase recognition sites stably integrated into them?

We apologize for any confusions in the manuscript. We have used HEK293T cells that does not have any artificially integrated recombinase recognition sites. This is clarified in our revised manuscript (lines 112-115).

3. The gene circuit schematics have several unlabeled features that are not annotated on the figure itself, or in the figure legends (e.g. recombinase sequences, promoters, etc.). Labeling these features and noting them in the legend will help the broader readership. Additionally, color code recognition sites in the schematics so that they match the genes which encode their recombinase. Finally, consider showing multiple steps in a schematic how each GPC is cleaved sequentially leading to activation of the next gene.

We thank the reviewer for improving our figures. We have revised figures 1b, 2a, 3b, 3d, 4a, Supplementary figures 1c, 2d, 2e, and the figure legends to clarify the schematics of the gene circuit.

4. The notation of the T in the schematics of the circuits is confusing. Typically, this is used as a transcriptional terminator in prokaryotic circuits (sequence that mark the end of transcription), but the authors use it to indicate the polyA tail. It is unclear why this needs special annotation – polyA tails provide stability to the RNA molecule and don't impact transcription. If the authors want to keep it in their schematics, it would be prudent to explain why it is necessary.

We thank the reviewer for pointing this out. We changed the "T" symbol for the polyadenylation signal to "poly A (pA)" in a white rectangle in figures 1a, 1b, 2a, 3b, 3d, 4a, Supplementary figures 1c, 2d, and 2e.

5. In figure 1b it would be useful to know the size between the red primers, especially since data is collected to determine whether the space between these primers is removed after recombination.

We have newly added the expected size of the PCR amplicon. As seen in revised figure 1e, the expected PCR amplicon is way too long to be efficiently amplified before excision.

6. The flow cytometry data in figure 1c is missing numbers on the axes. Despite being arbitrary values, it is always informative to know the difference between the lowest and highest values. Are the axes

linear? Log scale?

We revised our figure 1c and all other FACS plots (figures 2b, supplementary figures 2a, 2f) to indicate numbers. The axes were bi-exponential with the width basis modified to better monitor cells with low fluorescence for all flow cytometry plots.

7. The y-axes on these plots should be altered so the spread in the data around zero is larger. This can be done with flow data in FlowJo by changing the y-axis values to biexponential, then changing the width basis. This can also be done for Figure 2b and the flow data in the Extended Data.

We thank the reviewer for pointing this out. We revised our figures in line with the reviewer's point in figures 1c, 2b, Supplementary figures 2a and 2f.

8. Figure 1d shows a quantification of the flow cytometry data presented in 1c, however without a description of what cells are being looked at in the figure legend, this figure is a little confusing.

We replaced the label for the y axis in figure 1d to better explain the data. The recombination frequency is calculated as %BFP+/ (%BFP+ or GFP+) because BFP+ indicates successful recombination, and expression of either GFP or BFP means the cells are transfected. This is also newly explained in the corresponding figure legend. We apologize for any confusion in the previous version.

9. Figure 1e is not informative at all. Without knowing what size of the fragments before excision and no ladder present on the gel, this gel could be just about anything.

We replaced figure 1e with the new gel image to indicate the ladder. We found indicating the size of each band for the ladder quite distracting, and provided this in the unmodified gel image in Supplementary Fig. 6.

10. The same comments made for figure 1c apply to figure 2b.

We revised our figure 2b to indicate numbers. The axes were bi-exponential with width basis modified to better monitor cells with low fluorescence for all flow cytometry plots.

11. In 2C and 2D, the pie charts are quite large and are distracting (and they all look the same). Instead of the pie charts, consider preparing a time course graph (like was done with mean fluorescence) showing the average fraction of cells at each time point expressing each of the 4 fluorescent factors.

We thank the reviewer for this helpful suggestion. We replaced the pie charts to a time course graph in line with the reviewer's comments in figure 2c and d.

12. In figure 3B the authors swap out fluorescence reporter genes with other genes. It is never discussed why these particular genes were chosen. Indeed, in the main text it describes using the circuit for a cascade of endogenous genes RHOXF2, ASCL1, HBG1, and TTN, however, there is no indication that these have any natural relationship *in vivo*. It would be helpful if the authors could provide their reasoning for choosing these genes in particular to be expressed sequentially.

We thank the reviewer for pointing this out. The endogenous genes used for our proof-of-concept gene activation cascade were genes with well validated activation by CRISPRa. This is newly noted in our revised main text (lines 198-199).

13. Figure 3d should have an associated line graph, much like Figure 3c.

The quantification for figure 3d was in figure 3f. We replaced figure 3f as suggested in the reviewer's comment #15 (see below).

14. The gel in figure 3E needs a ladder. Also, what are the arrows supposed to be pointing to? This should be in the figure legend.

We replaced figure 3e to have a ladder in the figure. The arrows indicated the bands cleaved by T7 endonuclease. The expected size of the T7 endonuclease-cleaved fragment are newly indicated in figure 3e. The explanation for the arrows was also added in figure legends.

15. Figure 3f is not clear. A better explanation of this data in the legend and text would be very helpful. Why is APC expressed in all four stages? Why is there a lag in the data for MLH1 between stage 2 and stage 3? Finally, please consider displaying the data with the stages on the x-axis and use color to label the genes. This seems more intuitive. The same suggestion is for 4d, and 4e (as shown

below).

We thank the reviewer for this helpful suggestion. We have replaced our figures 3f and 4d, 4e as the reviewer suggested.

The payload gene in the first GPC is expressed in the absence of any CID ligand. Therefore, the sgRNA for the APC gene is already expressed at stage 1 and for this reason is mutated for all subsequent stages. The lag in the data for MLH1 between stages 2 and 3 can be explained by sgRNA transcribed at stage 2 remaining even after excision of the stage 2 GPC. The sgRNA expressed at stage 2 will remain active and form indels at the MLH1 locus until it gets degraded. We added this explanation in lines 217-221.

16. In Figure 4, the activation of GIB-PhiC removed the entire gene circuit in only 67% of the cells, leaving a fairly substantial 33% of cells with the circuit intact. This level of efficiency could certainly pose issues in any future translational work, if the goal was to completely remove the entire circuit. It would be nice if the authors could provide potential ways of improving the design to improve the removal of the entire circuit at the end, perhaps in the final concluding paragraph.

We thank the reviewer for pointing this out. We have performed the qPCR analysis to quantify gene circuit removal again, for the reason to be discussed in reviewer's point #18. The newly quantified gene circuit removal efficiency was 78%, higher than what was reported in the original manuscript. There were no significant changes in gene circuit amount in cells before gene circuit removal. However, we fully agree with the reviewer that there is significant room for improvement in the efficiency of gene circuit removal. We believe that the key to improving the gene expression cascade is optimizing the split recombinase activity by identifying the optimal split sites of the recombinases. This optimization process will involve making computational predictions of effective split sites for the recombinases and empirically validating the predictions. This is newly discussed in the new manuscript (line 264-277).

17. In 4b. Stage 3=1 does not make sense. Just say in the figure legend that you normalized to stage 3.

We apologize for the confusion. We changed the figure and figure legends as per the reviewer's comments.

18. How is the data in figure 4f analyzed? Delta delta Ct? Since it's relative gene content on the y-axis, it's unclear what exactly happening in this figure and the legend has no information on this. It also doesn't make sense that the circuit content increases going from stage 1 to 2. How can the circuit content increase with a removal of DNA? More description on this data would be very helpful.

We thank the reviewer for pointing this out. The data is analyzed by the ddCt method, as the reviewer thought. The excision efficiency is calculated as the relative amount of Cas9 DNA (which is supposed to be excised) normalized with Inverted Terminal Repeat (ITR) of the Piggybac transposon. There was one outlier in the quantification of the Cas9 DNA at stage 1. Because all other samples were normalized to the Cas9 DNA amount at stage 1, the outlier at stage 1 caused inaccuracy in the Cas9 DNA amount overall. We performed the qPCR analysis again with the identical samples, and replaced the original figure 4f with our new data. As shown in our new figure 4f, the amount of Cas9 DNA remained largely constant until stage 3B. The newly quantified excision efficiency was 78%. We have explained the gene circuit quantification scheme more thoroughly in the new figure legend and the methods section.

Minor comments:

1. Some full-forms of acronyms should be mentioned in the main text or in the figure legend before using it in the figures. For example: BFP, EFSp as used in Fig. 1b, CAGp as used in Fig. 2a, p2A.

We more carefully explained the full names of acronyms throughout the main text and figure legends, as per the reviewer's suggestions.

2. Since the entire paper uses split recombinase to a great extent, it is suggested that it is explained better to explain better in the introduction, along with its general mode of action of recombinase in a few lines.

We thank the reviewer for excellent suggestions. We added introductions for the split-recombinases in lines 58-64.

3. Extended Fig. 1c needs better explanation. What does the white bar stand for? They also did not

mention the polyadenylation signal that was finally selected after comparison. One suggested experiment is to test ligand treatment schedule for 24 hours and 36 hours with a 12-hour gap between switching of ligands. This would enable them to demonstrate the comparison of results between 24, 36 and 48 hours.

The white bar stands for the mCherry fluorescence in the absence of any polyadenylation signal. The BGH polyadenylation signal was chosen as the most leak-tight polyadenylation signal. This is now noted in lines 96-97. In Supplementary figure 2a, we have shown that extended ligand treatment (>24H) does not cause additional excision. Also in supplementary figure 2f, we have shown that extended gaps of time between switching of ligands do not significantly change the ability of cells to proceed to the next stage in the cascade when the next ligand is treated. These data suggest that the changes in the ligand treatment schedule will not significantly change the performance of the gene expression cascade.

4. In Fig. 1b, a split recombinase site can be shown in a black rectangle near BFP just like it has been shown for GFP for clarity of the concept

The GPCs tested in figure 1 had only one split recombinase appended to GFP. BFP had no split recombinase attached to it. We hope it clarifies reviewer's point.

5. Line 103: The word 'ligand treatment' should be removed to avoid repetition and for sentence accuracy.

Corrected. Thank you for pointing this out.

6. The colour code of the genes shown in Fig. 3b and Fig. 3d can be changed from those used for proteins shown in Fig. 2a to avoid confusion among readers

We thank the reviewer for helpful comments. We changed the color codes of the genes in figures 3b, 3d and 4a.

7. Line 172: There is an error in the spelling of the word 'knockout'.

Corrected. Thank you for pointing this out.

8. What does the x-axis on the graphs of Extended Fig. 1a stand for?

The numbers indicate the site the recombinases were split at, which is now explained in the figure legend. We apologize for any confusion.

Reviewers' Comments:

Reviewer #1:

Remarks to the Author:

The authors have addressed all my concerns. I have no more comment.

Reviewer #2:

Remarks to the Author:

The authors did an excellent job addressing all of my concerns. I recommend publication.